# Estimation of the Confidence Interval for the Ratio of the Coefficients of Variation of Two Weibull Distributions and Its Application to Wind Speed Data

**Manussaya La-ongkaew , Sa-Aat Niwitpong \* and Suparat Niwitpong**

Department of Applied Statistics, Faculty of Applied Science, King Mongkut's University of Technology North Bangkok, Bangkok 10800, Thailand
\* Correspondence: sa-aat.n@sci.kmutnb.ac.th

**Abstract:** The Weibull distribution, one of the most significant distributions with applications in numerous fields, is associated with numerous distributions such as generalized gamma distribution, exponential distribution, and Rayleigh distribution, which are asymmetric. Nevertheless, it shares a close relationship with a normal distribution where a process of transformation allows them to become symmetric. The Weibull distribution is commonly used to study the failure of components and phenomena. It has been applied to a variety of scenarios, including failure time, claims amount, unemployment duration, survival time, and especially wind speed data. A suitable area for installing a wind turbine requires a wind speed that is both sufficiently high and consistent, and so comparing the variation in wind speed in two areas is eminently desirable. In this paper, methods to estimate the confidence interval for the ratio of the coefficients of variation of two Weibull distributions are proposed and applied to compare the variation in wind speed in two areas. The methods are the generalized confidence interval (GCI), the method of variance estimates recovery (MOVER), and Bayesian methods based on the gamma and uniform priors. The Bayesian methods comprise the equal-tailed confidence interval and the highest posterior density (HPD) interval. The effectiveness of the methods was evaluated in terms of their coverage probabilities and expected lengths and also empirically applied to wind speed datasets from two different areas in Thailand. The results indicate that the HPD interval based on the uniform prior outperformed the others in most of the scenarios tested and so it is suggested for estimating the confidence interval for the ratio of the coefficients of variation of two Weibull distributions.

**Keywords:** coefficient of variation; Weibull distribution; Bayesian; generalized confidence interval; method of variance estimates recovery

## 1. Introduction

In statistics, Weibull distribution is a continuous probability distribution that is positively asymmetric. It is related to several other probability distributions, for instance, it interpolates between the exponential distribution and the Rayleigh distribution. In addition, we can apply a simple close-to-normal approximation of a Weibull random variable. Suppose that random variable $X$ follows a two-parameter Weibull distribution $Weibull(a, k)$, with $a$ as the scale parameter and $k$ as the shape parameter, then the probability density function (pdf) of $X$ can be defined as

$$f(x; a, k) = \frac{k}{a}\left(\frac{x}{a}\right)^{k-1} \exp\left[-\left(\frac{x}{a}\right)^k\right], x > 0, a > 0, k > 0, \tag{1}$$

where $E(X) = \mu = a\Gamma(1 + \frac{1}{k})$ and $V(X) = \sigma^2 = a^2[\Gamma(1 + \frac{2}{k}) - (\Gamma(1 + \frac{1}{k}))^2]$. Kulkarni and Powar [1] proposed transformation $Y = X^p$, where the power $p$ is chosen so that the distribution of transformed variable $Y$ only has very a small deviation from symmetry, and

simultaneously has tail behavior very close to that of normal distribution with the same mean and variance. To approximate the distribution of $Y$ to a normal distribution, $p = k\theta$ is exactly symmetric, where the value of $\theta$ is the solution of skewness equation $(t(\theta))$ of the distribution of $Y$ as follows:

$$t(\theta) = t(p,k) = \frac{\Gamma(1+3\theta) - 3\Gamma(1+\theta)\Gamma(1+2\theta) + 2[\Gamma(1+\theta)]^3}{[\Gamma(1+2\theta) - (\Gamma(1+\theta))^2]^{3/2}}. \tag{2}$$

The skewness function of $X$ is obtained by substituting $\mu$, $\sigma^2$, $p = k\theta$, and thus is no longer depends on scale parameter $a$. The Weibull distribution has been used in many fields, including engineering, industry, insurance claims and weather forecasting. For example, it has been used to analyze the survival time of guinea pigs injected with different doses of tubercle bacilli [2] and the lifetime the front and rear brake pads [3]. Xu et al. [4] analysed the effect of laser treatment in delay the onset of blindness in patients with diabetic retinopathy. Wang et al. [5] investigated inverse estimators for parameters of Weibull distribution and applied to the data on times to breakdown of an insulating fluid. Zhang et al. [6] studied the reliability estimation of the multicomponent stress-strength model involving one stress and two correlated strength components from a parallel system. Zhuang et al. [7] analyzed the progressive-stress accelerated life tests with group effect under progressive censoring. Pang et al. [8] approximated the Weibull distribution's parameters using wind speed data from a Hong Kong observatory. Yingni et al. [9] estimated the wind energy potential of 15 wind farms in China using the Weibull distribution. It has also been to estimate the distance traveled by a vehicle before throttle failure [10–12]. One interesting data consideration about Weibull distribution is wind speed data. In Thailand, a number of research studies have been conducted on the potential of wind energy in order to find suitable sites for wind turbine installations [13–18].

Wind is a significant source of renewable energy for the electricity generation that is clean and ecologically friendly. Over the years, Thailand has been unable to utilize wind power as efficiently as it should due to the high cost of building wind turbines. Therefore, selecting suitable sites for the installation of wind turbines and wind potential measurement stations is very important. If an area has poor wind speed potential, a station may have to be dismantled and installed at a new site with better wind energy potential. Sufficient wind power should have wind speeds that are not too low and consistent throughout the year, and the mean of the distribution of the wind speed data should be applied for statistical inference. However, since the distribution of the data is skewed, the mean may not be the best measure of the central tendency as it is very sensitive to extreme values in a small sample. Instead, the dispersion in the wind speed data over time is a better measure, with low dispersion being preferable.

The coefficient of variation (CV) is a measure of the degree of dispersion in a distribution. It is the ratio of the standard deviation to the mean, and, unlike the variance and standard deviation, the measurement unit of the original data is not involved, which makes it very useful for comparing the dispersion in multiple datasets with different units or very different means. Utilization of the CV is widespread, including in the fields of science, engineering, and medicine. For example, Billings et al. [19] estimated the CV for the impact of socioeconomic status on paying hospital bills. Kim et al. [20] analyzed variations in the cycle of hydrogen-fueled engines. Romano et al. [21] analyzed the shear and tensile bond strength of the tooth structure. Saelee et al. [22] examined the variability of agricultural production using an approximate confidence interval for the CV. Ospina and Marmolejo-Ramos [23] studied the stability of a robust CV estimator for psychological and genetic data. Estimating the CV can be achieved via point or interval estimation. Of the two, the confidence interval is more meaningful and provides better information on the parameter of interest than a point estimator [24]. For example, Yosboonruang et al. [25] estimated the confidence interval for the CV of rainfall data from Songkhla province in Thailand, while La-ongkaew et al. [26] estimated the confidence for the difference in the CVs of wind datasets from Trad and Chonburi provinces in Thailand. Either the difference

between or the ratio of the parameter of interest values can be applied to compare two populations. However, if the difference between the CVs is small, the conclusions drawn for the statistical inference may not be accurate. Therefore, the ratio of the parameter values is usually more appropriate than the difference between them. In the present study, we are interested in comparing the dispersion of wind speed data in two locations in Thailand. Since the dispersion of wind speed in a similar area or province may not be that different, we change constructed estimators for the confidence interval for the ratio of the CVs of two wind speed datasets. Estimating the confidence interval for the ratio of CVs of two populations has been studied in many instances.

For example, Verrill and Johnson [27] introduced the confidence bounds for the ratio of the CVs of normal distributions using both asymptotic and simulation procedures. Buntao and Niwitpong [28] estimated the confidence bounds for the ratio of the CVs of two independent delta-lognormal distributions using the concepts of the generalized variable approach (GVA) and the method of variance estimates recovery (MOVER). Sangnawakij et al. [29] provided two new estimators based on MOVER, Wald, and Score intervals for the confidence interval of the ratio of CVs of gamma distributions. Niwitpong and WongKhao [30] provided estimators for the confidence bounds for the ratio of the CVs of normal distributions with known ratio of variances based on GVA and MOVER. Moreover, estimating the confidence interval for the ratio of the CVs of two two-parameter exponential distributions was achieved using MOVER and the generalized confidence interval (GCI) method by Sangnawakij et al. [31]. Hasan and Krishnamoorthy [32] improved the confidence interval estimation for the ratio of the CVs of two lognormal distributions based on MOVER and the fiducial approach. Based on the GCI, Puggard et al. [33] estimated the confidence interval for the ratio of the CVs of Birnbaum–Saunders distributions and compared it along with the biased-corrected percentile bootstrap and the biased-corrected and accelerated approaches. Using the ideas of GCI and MOVER, Yosboonruang and Niwitpong [34] proposed new confidence interval estimator for the ratio of the CVs of delta-lognormal distributions. However, to the best of our knowledge, estimating the confidence interval for the ratio of the CVs of two Weibull distributions has not previously been considered.

Herein, we propose estimators for the confidence interval of the ratio of the CVs of two two-parameter Weibull distributions. The methods used are GCI, MOVER based on Hendricks and Robey's confidence interval, and Bayesian methods using the gamma and uniform priors, the details of which are given in Section 2. The details of a simulation study and the results thereof are covered in Section 3. Application of the methods to two wind speed datasets to illustrate their efficacy is provided in Section 4. Finally, a discussion and conclusions are presented in the last section.

## 2. Methods

Suppose that $X_i = (X_{i1}, X_{i2}, \ldots, X_{in_i}), i = 1, 2$ are independent two-parameter Weibull random variables, denoted as $Weibull(a_i, k_i)$, where positive constants $a_i$ and $k_i$ are the scale parameters and shape parameters, respectively. The pdf of $X_i$ is given by

$$f(x_{ij}; a_i, k_i) = \frac{k_i}{a_i} \left(\frac{x_{ij}}{a_i}\right)^{k_i - 1} \exp\left[-\left(\frac{x_{ij}}{a_i}\right)^{k_i}\right], x_{ij} > 0. \tag{3}$$

The maximum likelihood estimation can be used to estimate parameters $a_i$ and $k_i$. Since the maximum likelihood estimators lack a closed form, they must have been acquired numerically; (see the results of Cohen [35] and Lemon [36]). Maximum likelihood estimators $\hat{k}_i$ of $k_i$ can be obtained from the solution to the following equation:

$$\frac{1}{\hat{k}_i} - \frac{\sum\left[x_{ij}^{\hat{k}_i} \ln(x_{ij})\right]}{\sum(x_{ij}^{\hat{k}_i})} + \frac{1}{n_i} \sum \ln(x_{ij}) = 0. \tag{4}$$

Similarly, the maximum likelihood estimators of $\hat{a}_i$ of $a_i$ are defined by

$$\hat{a}_i = \left[ \sum x_{ij}^{\hat{k}_i} / n_i \right]^{\frac{1}{\hat{k}_i}}. \tag{5}$$

Furthermore, the mean, variance, and CV of $X_i$ are, respectively, obtained as

$$E(X_i) = \mu_i = a_i \Gamma \left( 1 + \frac{1}{k_i} \right) \tag{6}$$

$$V(X_i) = a_i^2 \left[ \Gamma \left( 1 + \frac{2}{k_i} \right) - \left( \Gamma \left( 1 + \frac{1}{k_i} \right) \right)^2 \right] \tag{7}$$

$$CV(X_i) = \lambda_i = \sqrt{ \frac{\Gamma \left( 1 + \frac{2}{k_i} \right)}{\left( \Gamma \left( 1 + \frac{1}{k_i} \right) \right)^2} - 1}. \tag{8}$$

Let $X_1 = (X_{11}, X_{12}, \ldots, X_{1n_1})$ and $X_2 = (X_{21}, X_{22}, \ldots, X_{2n_2})$ be random samples of size $n_1$ and $n_2$ from Weibull distributions with the parameters $a_1, a_2, k_1$ and $k_2$, respectively. Thus, the ratio of their CVs can be derived as

$$\beta = \frac{\lambda_1}{\lambda_2} = \frac{\sqrt{\dfrac{\Gamma \left( 1 + \frac{2}{k_1} \right)}{\left( \Gamma \left( 1 + \frac{1}{k_1} \right) \right)^2} - 1}}{\sqrt{\dfrac{\Gamma \left( 1 + \frac{2}{k_2} \right)}{\left( \Gamma \left( 1 + \frac{1}{k_2} \right) \right)^2} - 1}}. \tag{9}$$

Next, the six methods for estimating the confidence interval for $\beta$ are derived.

### 2.1. The GCI Approach

The important concept of GCI introduced by Weerahandi [37] uses the concept of the generalized pivotal quantity (GPQ).

Let $X = (X_1, X_2, \ldots, X_n)$ be a random variable from a distribution with the parameter $(\varphi, \gamma)$, where $\varphi$ is the parameter of interest and $\gamma$ is possibly a nuisance parameter, and $x$ is the observed value of $X$. The GPQ, $R(X; x, \varphi, \gamma)$ for confidence interval estimation, must satisfy the following conditions.

(GPQ1) The probability distribution of $R(X; x, \varphi, \gamma)$ is free of unknown parameters.

(GPQ2) The observed value of $R(X; x, \varphi, \gamma)$ at $X = x$ is the parameter of interest.

Afterward, the $100(1 - \alpha)\%$ confidence interval using the GCI for $\varphi$ is provided by $[R_\varphi(\alpha/2), R_\varphi(1 - \alpha/2)]$, where $R_\varphi(\alpha/2)$ is obtained by using $100(\alpha/2)$-th percentile of $R_\varphi(X; x)$.

The GPQs of the parameters of a Weibull distribution are presented by Krishnamoorthy et al. [38]. They presented that the distribution of $\frac{\hat{k}}{k}$ is $\hat{k}^*$ and the distribution of $\hat{k} \ln \frac{\hat{a}}{a}$ is $\hat{k}^* \ln \hat{a}^*$, neither of which depend on the parameter, and so they are the GPQs of $a$ and $k$. Let $\hat{a}^*$ and $\hat{k}^*$ be the maximum likelihood estimators from $Weibull(1, 1)$, and let $\hat{a}_0$ and $\hat{k}_0$ be the observed values of $\hat{a}$ and $\hat{k}$, respectively. Thus, the GPQs of the shape and scale parameters are, respectively, obtained by

$$R_k = \frac{k}{\hat{k}} \hat{k}_0 = \frac{\hat{k}_0}{\hat{k}^*}, \tag{10}$$

and

$$R_a = \left( \frac{a}{\hat{a}} \right)^{\frac{\hat{k}}{\hat{k}_0}} \hat{a}_0 = \left( \frac{1}{\hat{a}^*} \right)^{\frac{\hat{k}^*}{\hat{k}_0}} \hat{a}_0. \tag{11}$$

Suppose that $X_i = (X_{i1}, X_{i2}, \ldots, X_{in_i}), i = 1, 2$ are random samples of size $n_i$ from Weibull distributions, then the respective GPQs of the parameters can be defined as

$$R_{k_i} = \frac{k_i}{\hat{k}_i} \hat{k}_{i0} = \frac{\hat{k}_{i0}}{\hat{k}_i^*} , i = 1, 2, \tag{12}$$

and

$$R_{a_i} = \left(\frac{a_i}{\hat{a}_i}\right)^{\frac{\hat{k}_i}{\hat{k}_{i0}}} \hat{a}_{i0} = \left(\frac{1}{\hat{a}_i^*}\right)^{\frac{\hat{k}_i^*}{\hat{k}_{i0}}} \hat{a}_{i0} , i = 1, 2. \tag{13}$$

A useful feature of the GVA is that the GPQs of the functions of $a$ and $k$ can be obtained by simply plugging their GPQs into the function. The GPQ for function $g(a_i, k_i)$ is $g(R_{a_i}, R_{k_i})$. According to $\beta$ in Equation (9), which depends on shape parameter $k$, $R_\beta = \beta(R_{k_i})$. Thus, the GPQ of the ratio of the CVs of two Weibull distributions is given by

$$R_\beta = \frac{\sqrt{\dfrac{\Gamma\left(1 + \frac{2}{R_{k_1}}\right)}{\left(\Gamma\left(1 + \frac{1}{R_{k_1}}\right)\right)^2} - 1}}{\sqrt{\dfrac{\Gamma\left(1 + \frac{2}{R_{k_2}}\right)}{\left(\Gamma\left(1 + \frac{1}{R_{k_2}}\right)\right)^2} - 1}}. \tag{14}$$

Subsequently, the $100(1 - \alpha)\%$ two-sided GCI confidence interval for the ratio of the CVs of two Weibull distributions is

$$CI_{gci.\beta} = \left[L_{gci.\beta}, U_{gci.\beta}\right] = \left[R_\beta(\alpha/2), R_\beta(1 - \alpha/2)\right], \tag{15}$$

where $R_\beta(\alpha/2)$ is the $100(\alpha/2)$-th percentile of $R_\beta$.

### 2.2. The MOVER Approach

MOVER (Donner and Zou [39]) can be used to construct the confidence interval for the difference or the ratio of two distribution parameters. For the ratio of $\lambda_i, i = 1, 2$, the confidence interval is identified as

$$CI_m = [L_m, U_m], \tag{16}$$

where the lower bound and upper bound for $\hat{\lambda}_1/\hat{\lambda}_2$ are defined by

$$L_m = \frac{(\hat{\lambda}_1 \hat{\lambda}_2) - \sqrt{(\hat{\lambda}_1 \hat{\lambda}_2)^2 - l_1 u_2 (2\hat{\lambda}_1 - l_1)(2\hat{\lambda}_2 - u_2)}}{u_2 (2\hat{\lambda}_2 - u_2)}, \tag{17}$$

and

$$U_m = \frac{(\hat{\lambda}_1 \hat{\lambda}_2) + \sqrt{(\hat{\lambda}_1 \hat{\lambda}_2)^2 - u_1 l_2 (2\hat{\lambda}_1 - u_1)(2\hat{\lambda}_2 - l_2)}}{l_2 (2\hat{\lambda}_2 - l_2)}. \tag{18}$$

Suppose that $l_i$ and $u_i$ are the intervals of $\lambda_i$ (Hendricks and Robey [40]), then the confidence intervals for $\lambda_1$ and $\lambda_2$ can, respectively, be defined as

$$(l_1, u_1) = \left(\hat{\lambda}_1 - t_{(\alpha/2, n_1 - 1)} \frac{\hat{\lambda}_1}{\sqrt{2n_1}}, \hat{\lambda}_1 + t_{(\alpha/2, n_1 - 1)} \frac{\hat{\lambda}_1}{\sqrt{2n_1}}\right), \tag{19}$$

and

$$(l_2, u_2) = \left(\hat{\lambda}_2 - t_{(\alpha/2, n_2 - 1)} \frac{\hat{\lambda}_2}{\sqrt{2n_2}}, \hat{\lambda}_2 + t_{(\alpha/2, n_2 - 1)} \frac{\hat{\lambda}_2}{\sqrt{2n_2}}\right), \tag{20}$$

where $t_{(\alpha/2, n_1 - 1)}$ and $t_{(\alpha/2, n_2 - 1)}$ is obtained by using $100(\alpha/2)$-th percentiles of two t-distributions with degrees of freedom of $n_1 - 1$ and $n_2 - 1$, respectively.

Afterward, the $100(1 - \alpha)\%$ two-sided MOVER with Hendricks and Robey's confidence interval for the ratio of the CVs of two Weibull distributions is

$$CI_{mover.\beta} = \left[ L_{mover.\beta}, U_{mover.\beta} \right]. \tag{21}$$

### 2.3. The Bayesian Approaches

In Bayesian methodology, the posterior density is obtained from the posterior distribution as follows:

$$Posterior distribution \propto Prior distribution \times Likelihood function. \tag{22}$$

Suppose that $X$ is a random variable following a Weibull distribution, then another form of the pdf of $X$ is provided by

$$f(x; a', k) = a'kx^{k-1} \exp(a'x^k), x > 0, \tag{23}$$

where $a' = \left( \frac{1}{a} \right)^k$. In this study, we applied two priors for the parameters which are defined in the following subsections.

### 2.3.1. The Gamma Prior

First, we consider the independent priors of the two parameters from a Weibull distribution as follows:

$$\pi(k) \sim gamma(v_1, z_1), \tag{24}$$

and

$$\pi(a') \sim gamma(v_2, z_2), \tag{25}$$

where $v_1, z_1, v_2, z_2$ are the hyperparameters.

Accordingly, the joint posterior density function of $a'$ and $k$ given $x$ can be written as

$$\pi(a', k|x) \propto L(a', k|x) \times \pi(k)\pi(a'), \tag{26}$$

where $L(a', k|x)$ is a likelihood function.

Assuming that the priors in Equation (24) and Equation (25) are independent, then the conditional posterior distributions of parameters are, respectively, given by

$$\pi(k|a', x) \propto k^{n+v_1-1} \exp\left[ -kv_1 - a' \sum x^k \right], \tag{27}$$

and

$$\pi(a'|k, x) \sim gamma(n + v_2, z_2 + \sum x^k). \tag{28}$$

It can be seen that a sample of $a'$ can be obtained from a gamma distribution. However, the distribution of $\pi(k, a'|x)$ is not closed and for solving this problem, so that the Markov chain Monte Carlo (MCMC) (Geman and Geman [41]), a Gibbs sampling procedure was applied in the present study to generate a sample from the posterior density function. However, MCMC cannot be applied to the conditional posterior distribution of the shape parameter in a straightforward manner, so we combined it with the Random walk Metropolis (RWM) algorithm. The combined algorithm obtained using the Bayesian estimate for $\hat{\beta}^{(t)}$ is as follows.

1. Start with $(a'^{(0)}, k^{(0)})$, where it is an initial value

2. Generate $a'^{(t)}$ from gamma distribution with parameters $(n + v_2, z_2 + \sum x^{k^{(t-1)}})$

3. Update $k^{(t)}$ from RWM algorithm

   - Generate $\varepsilon$ from normal distribution with parameters $(0, \sigma_k^2)$
   - Calculate $k^*$ from $k^{(t-1)} + \varepsilon$
   - Calculate $A_k = \frac{L(k^*, a'|x)\pi(k^*)}{L(k, a'|x)\pi(k)}$
   - Generate variable $u$ from uniform distribution with parameters $(0, 1)$
   - If $u \leq min(1, A_k)$ set $k^{(t)} = k^*$, else set $k^{(t)} = k^{t-1}$

4. Calculate the parameter of interest, $\hat{\beta}^{(t)}$

5. Discard the first 1000 values of $\hat{\beta}^{(t)}$

For $i = 1, 2$, let $X_i = (X_{i1}, X_{i2}, \ldots, X_{in_i})$ be random samples from Weibull population with parameters $a_i$ and $k_i$. After we computed Bayesian estimates via the above algorithms, confidence interval estimation can be obtained from the percentile of the estimate as follows.

The $100(1 - \alpha)\%$ two-sided confidence interval for the ratio of CVs based on the Bayesian method using the gamma prior is given by

$$CI_{gamma.\beta} = \left[ L_{gamma.\beta}, U_{gamma.\beta} \right],\tag{29}$$

where $L_{gamma.\beta}$ and $U_{gamma.\beta}$ are the lower and upper bounds of the $100(1 - \alpha)\%$ equal-tailed confidence intervals and the highest posterior density (HPD) interval of $\beta$, respectively.

The HPD interval is the shortest interval in the HPD region when all of the values inside the HPD region have higher probability densities than any value outside of it [42]. The HPD interval was calculated by using the HDInterval package in the R programming suite.

2.3.2. Uniform Prior Distribution

The non-informative uniform prior can be applied as follows:

$$\pi(a) \sim uniform(0, 100)\tag{30}$$

and

$$\pi(k) \sim uniform(0, 4).\tag{31}$$

A sample from a joint posterior distribution can be acquired via Gibbs sampling. For a Weibull distribution, Khan and Ahmed [43] used the R2jags package in R programming (a Gibbs sampler) to summarize the posterior inference. Thereby, they specified the model for a two-parameter Weibull distribution and provided the code for generating an MCMC sample.

```
Model specification:
"model{
# Likelihood
for (i in 1:length(x)){
p[i] ← dweib(x[i],shape, theta);
ones[i] ∼ dbern(p[i]);
}
# Priors
shape ∼ dunif(0,4)
scale ∼ dunif(0,100)
theta ← pow(1/(scale), shape)
}", file = "weibullmodel.txt"),
```

where theta is a transformation of a scale parameter in another form. The R2jags functions and the arguments used for fitting the parameters of a Weibull distribution are obtained from Su and Yajima [44] as follows:

jags.fit ← jags(data, inits, parameters.to.save, n.iter = 20,000, model.file = "weibullmodel.txt", n.burnin = 1000)

From the sample obtained via R2jags (denoted as $\hat{\beta}^{(j)}$), the $100(1 - \alpha)\%$ Bayesian equal-tailed and the HPD interval based on uniform prior for the ratio of the CVs of two Weibull distributions are given by

$$CI_{uni.\beta} = \left[ L_{uni.\beta}, U_{uni.\beta} \right],\tag{32}$$

where $L_{uni.\beta}$ and $U_{uni.\beta}$ are the lower and upper bounds.

### 3. Simulation Results

Using the R statistical program, the performances of the confidence interval estimators were compared in terms of their coverage probability (CP) and expected length (EL). The best-performing method in each scenario had a CP greater than or equal to the nominal confidence level and the shortest EL.

For the simulation study, sample sizes $(n_1, n_2) = (10, 10)$, $(10, 30)$, $(10, 50)$, $(30, 30)$, $(30, 50)$, $(50, 50)$, $(50, 100)$, or $(100, 100)$; scale parameters $a_1 = a_2 = 2$; shape parameters $k_1 = 1$ and $k_2 = 0.5, 1, 2,$ or 4 with the ratio of the CVs of 0.4472, 1, 1.9130, or 3.5645, respectively. The number of times each situation is replicated was $M = 5000$, with $m = 2500$ for the GCI method. Furthermore, $T = 20,000$ realizations of MCMC were generated using the Gibbs algorithm with a burn-in of 1000. The nominal confidence level was set as 0.95.

The following algorithm 1 was used to obtain the CP and the EL of the confidence interval estimates.

---

**Algorithm 1:** The CP and EL of the confidence interval estimates for the ratio of the CVs

---

1. Set $M, m, T, n_1, n_2, a_1, a_2, k_1,$ and $k_2$
2. Generate $X_i = (X_{i1}, X_{i2}, \ldots, X_{in_i})$ from $Weibull(a_i, k_i)$
3. Construct generalized confidence interval $(L_{gci.\beta}, U_{gci.\beta})$ from Equation (15)
4. Construct MOVER confidence interval $(L_{mover.\beta}, U_{mover.\beta})$ from Equation (21)
5. Construct equal-tailed and HPD interval based on Bayesian using gamma prior distribution $(L_{gamma.\beta}, U_{gamma.\beta})$ from Equation (29)
6. Construct equal-tailed and HPD interval based on Bayesian using uniform prior distribution $(L_{uni.\beta}, U_{uni.\beta})$ from Equation (32)
   If $(L \le \beta \le U)$, then set $P = 1$, else set $P = 0$
7. Repeat steps 1–6 for $M$ times
8. Determine the average of $P$ for the CP
9. Determine the average of $(U - L)$ for the EL

---

The simulation results for $n_1 = n_2$ in Table 1, show that the GCI method yielded CP higher than or close to the nominal confidence level of 0.95 for all cases whereas those using the MOVER method were under estimated, except for $(n_1, n_2) = (10, 10)$. Of the Bayesian methods, the HPD interval based on the uniform prior outperformed the others for $k_2 = 1$ or 2 because its CPs were greater than the target. Furthermore, CPs of the HPD interval based on the gamma prior were over or close to 0.95 in most cases. Especially, for $(n_1, n_2) = (100, 100)$ and $k_2 = 0.5$, it also provided the shortest EL. The Bayesian equal-tailed confidence interval based on the gamma prior and uniform prior yielded CPs close to the goal, except for the Bayesian method based on the uniform prior when $k_2 = 0.5$ or 4. The results for $n_1 \ne n_2$ in Table 2 were similar to those for $n_1 = n_2$ in Table 1 in that the Bayesian methods based on the uniform prior provided CPs greater than 0.95 and the shortest ELs in most cases. The CPs obtained with the equal-tailed confidence interval satisfied the target for $(n_1, n_2) = (10, 30)$ and $(10, 50)$ and $k_2 = 1$ or 2; $(n_1, n_2) = (30, 50)$, and $k_2 = 2$. Meanwhile, the HPD interval obtained CPs that satisfied the target for $(n_1, n_2) = (30, 50)$ and $k_2 = 1$, and $(n_1, n_2) = (50, 100)$ and $k_2 = 1$ or 2. Finally, the CPs and ELs of the methods in Tables 1 and 2 are summarized in Figures 1 and 2.

**Table 1.** Performance indicators for the six approaches for the 95% confidence intervals for the ratio of CVs of Weibull distributions for $n_1 = n_2$.

| $(n_1, n_2)$ | $k_2$ | Coverage Probability (Expected Length) | | | | | |
|---|---|---|---|---|---|---|---|
| | | GCI | MOVER | Gamma | | Uniform | |
| | | | | Equal-Tailed | HPD | Equal-Tailed | HPD |
| (10, 10) | 0.5 | 0.9520 | 0.9656 | 0.9368 | 0.9416 | 0.9624 | **0.9642** |
| | | (2.0648) | (1.9351) | (1.8019) | (1.6623) | (1.5806) | **(1.4741)** |
| | 1 | 0.9504 | 0.9702 | 0.9376 | 0.9428 | 0.9616 | **0.9510** |
| | | (3.7588) | (3.6355) | (3.2665) | (2.9949) | (2.6755) | **(2.4894)** |
| | 2 | 0.9460 | 0.9730 | 0.9310 | 0.9390 | 0.9560 | **0.9541** |
| | | (3.7833) | (3.6691) | (3.3161) | (3.0380) | (2.6936) | **(2.5071)** |
| | 4 | 0.9498 | **0.9696** | 0.9152 | 0.9258 | 0.8808 | 0.8176 |
| | | (7.1887) | **(6.8958)** | (6.1165) | (5.5691) | (3.3775) | (3.1385) |
| (30, 30) | 0.5 | 0.9418 | 0.8738 | 0.9390 | 0.9406 | 0.0636 | 0.0806 |
| | | (0.4774) | (0.3689) | (0.4648) | (0.4553) | (0.4301) | (0.4216) |
| | 1 | 0.9466 | 0.9356 | 0.9438 | 0.9444 | 0.9666 | **0.9622** |
| | | (0.8855) | (0.8066) | (0.8590) | (0.8378) | (0.7359) | **(0.7215)** |
| | 2 | 0.9432 | 0.9414 | 0.9412 | 0.9456 | 0.9532 | **0.9560** |
| | | (1.6203) | (1.5406) | (1.5718) | (1.5290) | (1.3727) | **(1.3489)** |
| | 4 | 0.9426 | 0.9414 | 0.9298 | 0.9312 | 0.8830 | 0.8492 |
| | | (3.0110) | (2.8598) | (2.8841) | (2.8005) | (1.8310) | (1.8076) |
| (50, 50) | 0.5 | **0.9502** | 0.8632 | 0.9490 | 0.9466 | 0.0046 | 0.0074 |
| | | **(0.3509)** | (0.2665) | (0.3458) | (0.3413) | (0.3270) | (0.3232) |
| | 1 | 0.9486 | 0.9414 | 0.9436 | 0.9474 | 0.9568 | **0.9516** |
| | | (1.1872) | (1.1296) | (1.1657) | (1.1461) | (1.0505) | **(1.0390)** |
| | 2 | 0.9494 | 0.9446 | 0.9462 | 0.9474 | 0.9576 | **0.9506** |
| | | (1.1938) | (1.1354) | (1.1727) | (1.1528) | (1.0525) | **(1.0410)** |
| | 4 | 0.9474 | 0.9388 | 0.9392 | 0.9408 | 0.8928 | 0.8664 |
| | | (2.2261) | (2.1142) | (2.1671) | (2.1279) | (1.4584) | (1.4470) |
| (100, 100) | 0.5 | 0.9518 | 0.8604 | 0.9500 | **0.9504** | 0.0000 | 0.0000 |
| | | (0.2423) | (0.1821) | (0.2404) | **(0.2386)** | (0.2377) | (0.2359) |
| | 1 | 0.9478 | 0.9304 | 0.9474 | 0.9484 | 0.9632 | **0.9602** |
| | | (0.4471) | (0.4053) | (0.4429) | (0.4390) | (0.3988) | **(0.3955)** |
| | 2 | 0.9512 | 0.9414 | 0.9500 | 0.9484 | 0.9576 | **0.9518** |
| | | (0.8142) | (0.7749) | (0.8068) | (0.7990) | (0.7486) | **(0.7435)** |
| | 4 | **0.9516** | 0.9408 | 0.9450 | 0.9434 | 0.8980 | 0.8852 |
| | | **(1.5205)** | (1.4443) | (1.4908) | (1.4752) | (1.0810) | (1.0748) |

The CP higher than the target of 0.95 and the shortest EL are in bold.

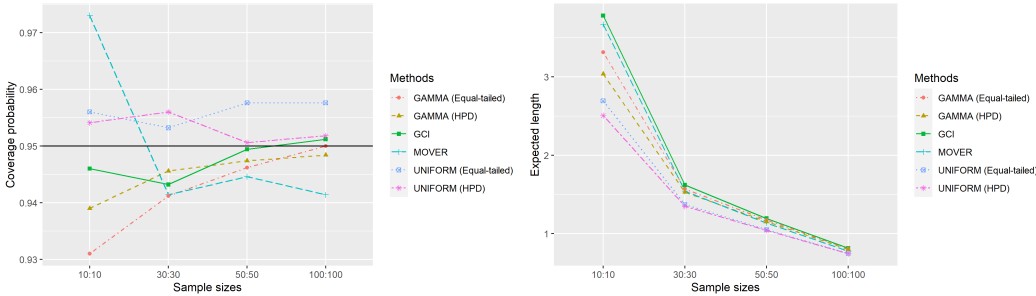

**Figure 1.** The performance of the six approaches in terms of CP and EL for equal sample sizes.

**Table 2.** Performance indicators for the six approaches for the 95% confidence intervals for the ratio of CVs of Weibull distributions for $n_1 \neq n_2$.

| $(n_1, n_2)$ | $k_2$ | Coverage Probability (Expected Length) | | | | | |
| --- | --- | --- | --- | --- | --- | --- | --- |
| | | GCI | MOVER | Gamma | | Uniform | |
| | | | | Equal-Tailed | HPD | Equal-Tailed | HPD |
| (10, 30) | 0.5 | 0.9486 | 0.9072 | 0.9402 | 0.9374 | 0.3928 | 0.5386 |
| | | (0.8166) | (0.5324) | (0.6863) | (0.6292) | (0.8269) | (0.7489) |
| | 1 | 0.9494 | 0.9146 | 0.9326 | 0.9248 | **0.9558** | 0.9386 |
| | | (1.6640) | (1.1570) | (1.3775) | (1.2518) | **(1.1762)** | (1.0756) |
| | 2 | 0.9486 | 0.9242 | 0.9368 | 0.9244 | **0.9508** | 0.9268 |
| | | (3.1679) | (2.2231) | (2.6069) | (2.3608) | **(2.1986)** | (2.0139) |
| | 4 | **0.9524** | 0.9286 | 0.9394 | 0.9294 | 0.9132 | 0.8626 |
| | | **(5.8946)** | (4.1368) | (4.8418) | (4.3767) | (3.2922) | (3.0077) |
| (10, 50) | 0.5 | 0.9514 | 0.8840 | 0.9376 | 0.9204 | 0.3404 | 0.4954 |
| | | (0.7584) | (0.4823) | (0.6248) | (0.5688) | (0.8235) | (0.7448) |
| | 1 | 0.9520 | 0.9078 | 0.9416 | 0.9300 | **0.9608** | 0.9402 |
| | | (1.6131) | (1.0692) | (1.3127) | (1.1862) | **(1.2534)** | (1.0941) |
| | 2 | 0.9500 | 0.9050 | 0.9344 | 0.9206 | **0.9500** | 0.9200 |
| | | (3.0181) | (2.0274) | (2.4529) | (2.2141) | **(2.3623)** | (2.0736) |
| | 4 | 0.9478 | 0.9034 | 0.9356 | 0.9216 | 0.9194 | 0.8700 |
| | | (5.6118) | (3.7664) | (4.5533) | (4.1062) | (3.3073) | (3.0073) |
| (30, 50) | 0.5 | 0.9512 | 0.8896 | 0.9476 | 0.9466 | 0.0294 | 0.0386 |
| | | (0.4070) | (0.3095) | (0.3940) | (0.3848) | (0.4112) | (0.4032) |
| | 1 | 0.9484 | 0.9312 | 0.9470 | 0.9490 | 0.9668 | **0.9618** |
| | | (0.7873) | (0.6847) | (0.7589) | (0.7381) | (0.6423) | **(0.6296)** |
| | 2 | 0.9518 | 0.9422 | 0.9482 | 0.9444 | **0.9574** | 0.9458 |
| | | (1.4532) | (1.3041) | (1.3967) | (1.3562) | **(1.2026)** | (1.1812) |
| | 4 | 0.9488 | 0.9394 | 0.9396 | 0.9392 | 0.8996 | 0.8706 |
| | | (2.7121) | (2.4311) | (2.5940) | (2.5152) | (1.7579) | (1.7306) |
| (50, 100) | 0.5 | 0.9470 | 0.8820 | 0.9464 | 0.9442 | 4e-04 | 8e-04 |
| | | (0.2894) | (0.2241) | (0.2839) | (0.2796) | (0.3228) | (0.3190) |
| | 1 | 0.9510 | 0.9282 | 0.949 | 0.9458 | 0.9656 | **0.9576** |
| | | (0.5648) | (0.4967) | (0.5523) | (0.5425) | (0.4780) | **(0.4723)** |
| | 2 | 0.9530 | 0.9444 | 0.9493 | 0.9491 | 0.9520 | **0.9500** |
| | | (1.0335) | (0.9400) | (1.0118) | (0.9923) | ( 0.9054) | **(0.8957)** |
| | 4 | **0.9600** | 0.9450 | 0.9465 | 0.9460 | 0.8909 | 0.8860 |
| | | **(1.8778)** | (1.7113) | (1.8156) | (1.7801) | (1.3977) | (1.3818) |

The CP higher than the target of 0.95 and the shortest EL be in bold.

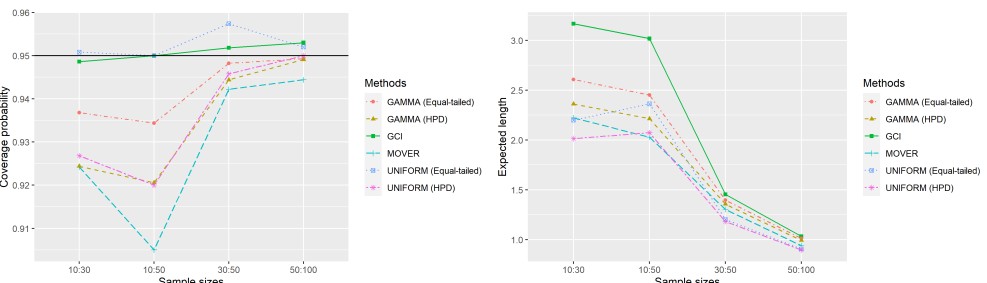

**Figure 2.** The performance of the six approaches in terms of CP and EL for unequal sample sizes.

## 4. Applications

### 4.1. Example 1

Data on wind speed collected in April and May 2019 by the Department of Alternative Energy Development and Efficiency, the Thai Ministry of Energy [45] from wind energy stations in Thailand's southern and central regions, were used to demonstrate the effectiveness of the proposed approaches. The Weibull Q-Q plots for the datasets with *p*-values

of 0.4442 and 0.6988, respectively, in Figure 3, and the Akaike Information Criterion (AIC) values in Table 3 confirm that the Weibull distribution is the best fits. The statistics of the datasets are $n_1 = 12, \hat{a}_1 = 2.5334, \hat{k}_1 = 2.3957, \hat{\mu}_1 = 2.2457, \hat{\sigma}^2_1 = 0.9964, \hat{\lambda}_1 = 0.4445$, and $n_2 = 6, \hat{a}_2 = 5.0390, \hat{k}_2 = 3.4741, \hat{\mu}_2 = 4.5320, \hat{\sigma}^2_2 = 2.0248, \hat{\lambda}_2 = 0.3186$. The ratio of CVs for the two Weibull distributions is $\hat{\beta} = 1.3954$. From the 95% confidence interval estimates based on all approaches as shown in Table 4, it is clearly that the length of the HPD interval using uniform prior was the shortest. Therefore, this result supports the simulation result for small sample sizes. Thus, the HPD interval using the uniform prior is suitable for estimating the confidence interval for the ratio of the CVs of the two wind speed datasets. Moreover, the interval between the lower and upper bounds of the confidence interval estimates for all of the proposed methods provide is 1. Thus, it is reasonable to conclude that the CVs of the two wind speed datasets are not significantly different.

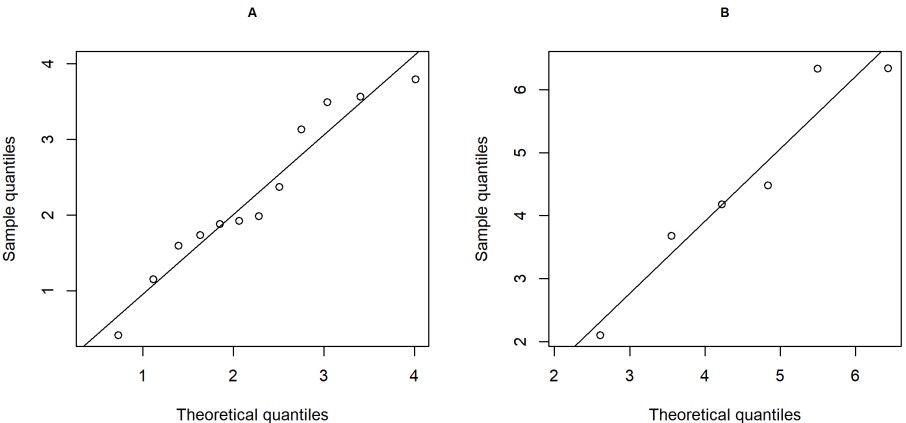

**Figure 3.** Weibull Q-Q plot of wind speed data in (**A**) southern (**B**) central.

**Table 3.** AIC values for fitting the five distributions.

|  | Distributions | | | | |
| --- | --- | --- | --- | --- | --- |
|  | **Weibull** | **Gamma** | **Log-Normal** | **Exponential** | **Normal** |
| Southern | **37.9176** | 39.2535 | 41.5439 | 45.4623 | 38.0888 |
| Central | **25.5902** | 26.1197 | 26.5604 | 32.0977 | 25.7886 |

The smallest AIC of the distribution be in bold.

**Table 4.** The 95% confidence intervals for the ratio of CVs based on two regions of wind speed data.

| Methods | Confidence Intervals for $\beta$ | | |
| --- | --- | --- | --- |
|  | **Lower** | **Upper** | **Length** |
| GCI | 0.5474 | 2.7031 | 2.1557 |
| MOVER | 0.6185 | 5.5925 | 4.9740 |
| Bayesian: Gamma prior (Equal-tailed) | 0.6204 | 2.6643 | 2.0439 |
| Bayesian: Gamma prior (HPD) | 0.5482 | 2.5195 | 1.9713 |
| Bayesian: Uniform prior (Equal-tailed) | 0.5686 | 2.1387 | 1.5701 |
| Bayesian: Uniform prior (HPD) | **0.4905** | **2.0174** | **1.5268** |

The shortest length of the proposed method be in bold.

### 4.2. Example 2

In this case, the first dataset comprises wind speed entries from stations on surface area provided by the Department of Energy Development and Promotion (DEDP), the Electricity Generating Authority of Thailand, and the Royal Thai Air Force while the second dataset comprises wind speed entries provided by stations in the sea and coastal areas situated on buoys belonging to the National Research Council of Thailand, lighthouses belonging to the Thai Royal Navy, gas platforms belonging to the Union Oil Company of California,

and belonging to the Defense Meteorological Satellite Program. This information was obtained from a report by the DEDP prepared by the Fellow Engineering Consultants Company Limited [46]. Weibull Q-Q plots in Figure 4 with *p*-values of 0.3988 and 0.0833, respectively, and the AIC values in Table 5 indicate that once again, the Weibull distribution provides the best fit for these datasets. The statistics for the two wind speed datasets are $n_1 = 24, \hat{a}_1 = 2.5736, \hat{k}_1 = 2.7015, \hat{\mu}_1 = 2.2887, \hat{\sigma}^2_1 = 0.8347, \hat{\lambda}_1 = 0.3992$, and $n_2 = 36, \hat{a}_2 = 4.3482, \hat{k}_2 = 5.0395, \hat{\mu}_2 = 3.9941, \hat{\sigma}^2_2 = 0.8249, \hat{\lambda}_2 = 0.2274$, while the ratio of their CVs $\hat{\beta} = 1.7555$. The 95% confidence intervals for $\beta$ are presented in Table 6. These findings suggest that the length of the HPD interval using uniform prior distribution was once again the shortest. According to the results in Table 6, the lower and the upper bounds of the confidence interval provide a length that is higher than 1, and thus it can be stated that the CV of the wind speed data from stations on surface area is significantly higher than that of the wind speed data from stations in the sea and coastal areas. This indicated that the potential of obtaining wind energy from sea and coastal areas is greater than obtaining it from wind turbines on surface area.

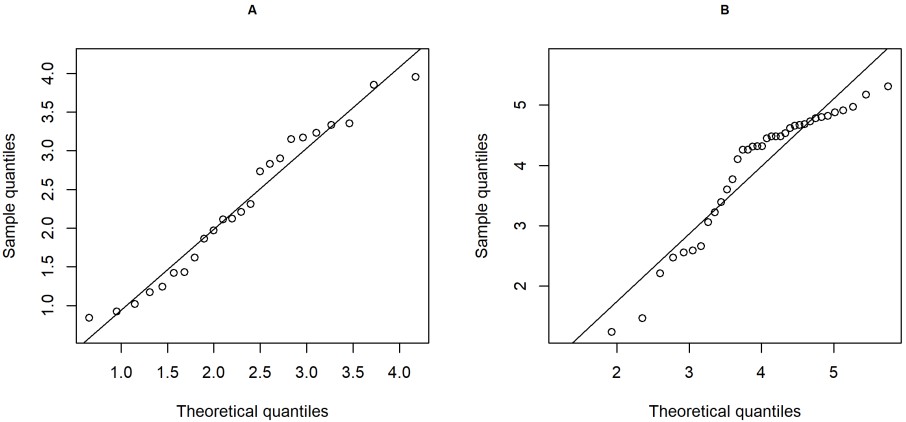

**Figure 4.** Weibull Q-Q plot wind speed data in (**A**) surface area (**B**) sea and coastal area.

**Table 5.** AIC values for fitting the five distributions.

|  | Distributions | | | | |
| --- | --- | --- | --- | --- | --- |
|  | **Weibull** | **Gamma** | **Log-Normal** | **Exponential** | **Normal** |
| Surface area | **67.4324** | 68.3245 | 69.5641 | 89.5692 | 68.7864 |
| Sea and coastal area | **105.1392** | 118.2662 | 124.4380 | 173.4727 | 108.9670 |

The smallest AIC of the distribution be in bold.

**Table 6.** The 95% confidence intervals for the ratio of CVs based on two areas of wind speed data.

| **Methods** | **Confidence Intervals for $\beta$** | | |
| --- | --- | --- | --- |
|  | **Lower** | **Upper** | **Length** |
| GCI | 1.3134 | 2.7317 | 1.4183 |
| MOVER | 1.2370 | 2.7271 | 1.4901 |
| Bayesian: Gamma prior (Equal-tailed) | 1.4087 | 2.5853 | 1.1767 |
| Bayesian: Gamma prior (HPD) | 1.3498 | 2.4569 | 1.0171 |
| Bayesian: Uniform prior (Equal-tailed) | 1.0129 | 1.8634 | 0.8505 |
| Bayesian: Uniform prior (HPD) | **1.0008** | **1.8244** | **0.8236** |

The shortest length of the proposed method be in bold.

## 5. Discussion

La-ongkaew et al. [26] proposed Bayesian methods based on the gamma prior for establishing the confidence interval for the difference between the CVs of a Weibull distribution, in which the Bayesian HPD interval performed the best in most cases. In the present

research, of the Bayesian methods based on the gamma and uniform priors for estimating the confidence interval for the ratio of the CVs of Weibull distributions, the HPD interval using the uniform prior performed well in most cases since its CPs were higher than or close to the target and it obtained the shortest ELs. Nevertheless, it is unsuitable when shape parameter $k_2 = 0.5$, which illustrates the limitation of using this method. Moreover, the ELs tended to decrease when the ratio of the CVs decreased, thereby showing its robustness. In addition, from the results of La-ongkaew et al. [26] found that the bootstrap method yielded the coverage probabilities highly lower than 0.95, which happened as well as constructing a confidence interval of the ratio of the CVs of two Weibull distributions. Thus, we did not consider using this approach in the present study.

## 6. Conclusions

GCI, MOVER, and Bayesian methods using the gamma and uniform priors were derived to estimate the confidence interval for the ratio of the CVs of two Weibull distributions. Simulation study results reveal that the Bayesian HPD interval using the uniform prior performed the best for shape parameter $k_2 = 1$ or 2 but not for $k_2 = 0.5$ or 4, thereby indicating the limitation of using this method. However, GCI and the Bayesian methods based on the gamma prior can be used in these scenarios instead.

**Author Contributions:** Conceptualization, S.-A.N.; Methodology, M.L.-o., S.-A.N. and S.N.; Software, M.L.-o.; Formal analysis, M.L.-o. and S.N.; Investigation, S.-A.N. and S.N.; Project administration, S.-A.N.; Resources, S.-A.N.; Data curation, M.L.-o.; Writing—original draft, M.L.-o.; Writing—review and editing, S.-A.N. and S.N.; Supervision, S.-A.N. and S.N. All authors have read and agreed to the published version of the manuscript.

**Funding:** This research was funded by King Mongkut's University of Technology North Bangkok (Grant No.KMUTNB-PHD-62-04)

**Data Availability Statement:** Data may be made available by contacting the corresponding author.

**Acknowledgments:** The authors are grateful to the Referees for taking the time and effort necessary to review the manuscript. We sincerely appreciate all valuable comments and suggestions, which helped us to improve the quality of the manuscript. We would like to express our gratitude to King Mongkut's University of Technology North Bangkok for funding our research and providing a venue for programming.

**Conflicts of Interest:** The authors declare no conflict of interest.

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
