# Peer review of "Estimation of the Confidence Interval for the Ratio of the Coefficients of Variation of Two Weibull Distributions and Its Application to Wind Speed Data"

_symmetry, doi:10.3390/sym15010046_

Round 1
Reviewer 1 Report
My comments are in the attached PDF file.

Author Response
Response to the Referee
Title: Confidence intervals for the ratio of coefficients of variation of Weibull distributions with applied to wind speed data
Authors: Manussaya La-ongkaew, Sa-Aat Niwitpong and Suparat Niwitpong
Dear referee,
We would like to express our gratitude to referees, who have given valuable suggestions and comments towards the improvement of our manuscript. This manuscript was revised according to the suggestions of referees. Here is one-by-one response to your comments as below. Please see attached.
Reviewer 1:
- In Section 1 (Pages 1-2), why is exactly symmetric?
Response: Based on the research of Kulkarni and Powar [1], they proposed a simple close-to-normal approximation to a Weibull random variable, which is “The Close-To-Normal Power Transformation”. They consider a transformation , where the power is chosen so that the distribution of the transformed variable has very small deviation from symmetry and simultaneously has tail behavior very close to that of the normal distribution with the same mean and variance. From Equation (2), they find a solution for that obtained , which the distribution of the variable where is exactly symmetric. The second key, to make the tail behavior of the distribution of very close that of normal distribution with the same mean and standard deviation as that of , they solve the equation for finding and , which are 0.2698 and 0.2994. To control the symmetry and tail behavior of distribution of transformed simultaneously close to the normal distribution, they suggest taking , where .
- What is in Equation (2)?
Response: Thank you for pointing out our mistake. We edited it to “”, which is the shape parameter of Weibull distribution.
- It is interesting that Equation (2) has nothing to do with a scale parameter . If possible, detail Equation (2).
Response: The Equation (2) is a skewness function, which does not depend on a scale parameter . Sine, the formula of the skewness is given by
where mean and variance . We substitute and in above equation, then they do not have scale parameter . Let (that means ), the above equation becomes Equation (2). To better understand, we have further explained Equation 2 in the manuscript.
- Line 134 on Page 4. If is used, do we have to estimate parameters under the assumption of the Weibull distribution? It is known that is an exponential distribution.
Response: Thank you for pointing out our mistake. We have explained as follows. The and are the maximum likelihood estimator based on sample from , which is one of the components for constructing the GPQ of and . Previously, we did not fully describe the details, which have already been edited.
- Equation on Page 5. What is the distribution of ? We need to know the distribution of to obtain the
Response: The GPQ (R) is a function that correspond two conditions. Firstly, we edited the details of constructing the GPQ of and in revised manuscript. From Krisnamoothy (2009), they showed that the distribution of is and the distribution of is , which do not depend on and . Hence, they are GPQ of and . A useful feature of the generalized variable approach is that a GPQ for a function and can be obtained by simply plugging their GPQ in the function. A GPQ for a function is given . According to in Equation (9) which depend on shape parameter . Therefore . To better understand, we have further explained in the revised manuscript.
- Pages 9 and 10. In Tables 1 and 2, Gci should read GCI, and Mover should read MOVER.
Response: Thank you for your suggestion. We already changed them.
- Page 11. In the applications, it would be interesting or better if the p-value for the Weibull Q-Q plot is reported. One way is to use weibullness R package to calculate the p-value, which is based on the Weibull Q-Q plot.
Response: Thank you for your suggestion. We already reported p-value for the Weibull Q-Q plot using “weibullness” R package.
- In Tables 1 and 2, the GCI method seems to outperform the HPD method especially for a smaller value of . Also, it looks like that the coverage probabilities of the GCI method are overall close to the target value of 0.95.
Response: To choose a recommended method, there are two important considerations: coverage probability and expected length. We first considered coverage probability values greater than or at least equal to 0.95, for which the chosen method must also provide the shortest expected length.

Reviewer 2 Report
The paper is quite clear and organized well. Several methods are used to construct confidence intervals of the ratio of coefficients of variation (CV) based on Weibull distributions. However, it can be improved as follows.
1. In lines 31-52, the authors introduces some backgrounds of the paper. However, I do not get the importance of CV in this example. Usually, CV is used as a variation indicator. Why do you use CV rather than mean or median in the example?
2. In subsection 2.1, for generalized inference method, Wang et al. (2010) have also proposed a generalized inference algorithm for Weibull distribution. What’s the difference between the proposed method and Wang et al. (2010).
Wang B, Yu K, Jones M. Inference under progressively type II right-censored sampling for certain lifetime distributions. Technometrics 2010;52(4):453–60.
3. In subsection 2.3, the authors used two kinds of priors: gamma priors and uniform prior. What’s the benefits of using gamma priors? In some references, for example, Bayesian inference of system reliability for multicomponent stress-strength model under Marshall-Olkin Weibull distribution. Systems 2022; 10(6):196. https://doi.org/10.3390/systems10060196, semi-conjugate priors are used and have good performance. For uniform prior, how to determine upper bounds? Any guidelines for this? In some cases, the upper bounds 100 and 4 in equations (30) and (31) may be not suitable.
4. In section 4, explain the results clearly in the two examples. For example, you get the estimate of CV, what does this value mean in the examples? How to explain the CV related to the practical background of data.
5. Some related works on Weibull distributions should be reviewed, such as “A unified model for system reliability evaluation under dynamic operating conditions”,”Modelling and estimation of system reliability under dynamic operating environments and lifetime ordering constraints”
Author Response
Response to the Referee
Title: Confidence intervals for the ratio of coefficients of variation of Weibull distributions with applied to wind speed data
Authors: Manussaya La-ongkaew, Sa-Aat Niwitpong and Suparat Niwitpong
Dear referee,
We would like to express our gratitude to referees, who have given valuable suggestions and comments towards the improvement of our manuscript. This manuscript was revised according to the suggestions of referees. Here is one-by-one response to your comments as below.
Reviewer 2:
The paper is quite clear and organized well. Several methods are used to construct confidence intervals of the ratio of coefficients of variation (CV) based on Weibull distributions. However, it can be improved as follows.
- In lines 31-52, the authors introduce some backgrounds of the paper. However, I do not get the importance of CV in this example. Usually, CV is used as a variation indicator. Why do you use CV rather than mean or median in the example?
Response: Thank you for your suggestion. In lines 31-52, We intend to explain about the importance of the dispersion of wind speed. Nevertheless, a long dense paragraph does not make sense of this. Thus, we already improved it. The importance of the of CV have described in next paragraph.
- In subsection 2.1, for generalized inference method, Wang et al. (2010) have also proposed a generalized inference algorithm for Weibull distribution. What’s the difference between the proposed method and Wang et al. (2010).
“Wang B, Yu K, Jones M. Inference under progressively type II right-censored sampling for certain lifetime distributions. Technometrics 2010;52(4):453–60.”
Response: The principle of generating the GPQ of the parameters of interest is similar, which correspond two conditions: GPQ1 and GPQ2 in section 2.1. Previously, we did not show enough details about constructing GPQ of our parameters of interest. Nevertheless, we have already improved it. Our research focuses on the CV of the Weibull distributed data, and the research of Wang et al. focuses on the shape, scale, and other quantities value of data that includes Type II censored data observations. Therefore, the formula of GPQs is different.
- In subsection 2.3, the authors used two kinds of priors: gamma priors and uniform prior. What’s the benefits of using gamma priors? In some references, for example, Bayesian inference of system reliability for multicomponent stress-strength model under Marshall-Olkin Weibull distribution. Systems 2022; 10(6):196. https://doi.org/10.3390/systems 10060196, semi-conjugate priors are used and have good performance. For uniform prior, how to determine upper bounds? Any guidelines for this? In some cases, the upper bounds 100 and 4 in equations (30) and (31) may be not suitable.
Response: From our literature reviews, gamma prior is an efficient alternative for Bayesian estimation for the Weibull distribution, which is widely used. For example, Gupta and Singh studied the classical and Bayesian estimation of Weibull distribution; they results indicated that Bayes estimation with gamma prior provided more precise estimates as compared with Jeffrey’s prior and maximum likelihood estimates. Saraiva and Suzuki studied the Bayesian methods for estimation of parameters Weibull distribution in presence of right-censored data; they found that the Bayesian with gamma prior using random walk metropolis outperforms the maximum likelihood estimation. And from the results of Mondal and Kundu, the Bayes estimators of parameters are better than the maximum likelihood estimators. For uniform prior, this was experimentally performed on other parameters and it was found that 100 and 4 were good values, which in accordance with the results of Khan and Ahmed.
- In section 4, explain the results clearly in the two examples. For example, you get the estimate of CV, what does this value mean in the examples? How to explain the CV related to the practical background of data.
Response: Thank you for your suggestion. We have explained about wind speed data in section “Application”. The summary statistics; sample mean, variance, and CV have been successfully added. In addition, we expanded on the results of two examples in the revised manuscript.
- Some related works on Weibull distributions should be reviewed, such as “A unified model for system reliability evaluation under dynamic operating conditions”, “Modelling and estimation of system reliability under dynamic operating environments and lifetime ordering constraints”
Response: Thank you for your suggestion. We already added the related works in section “Introduction”.

Reviewer 3 Report
the English is uniformly poor and must be improved by careful editing by a native speaker. Almost every sentence contains a grammatical error. The meaning is usually (but not always) clear but the poorly structured sentences give a very amateurish feel to the manuscript.
The authors seem to confuse global warming with the catalytic degradationd of the ozone layer: these are very different processes and the authors seem to be quite poorly informed about both of them. The long dense paragraphs on pages 2 and 3 are out of place in an introduction.
It is not clear to me in what way the CV is superior as an inferential tool to the variance and the authors do not make this clear. I would expect a self-contained discussion of this issue. Why not consider a ratio of variances which would have a notional Fisher distribution]? What exactly goes wrong by this that considering ratios of CVs ameliorates?
p2line104 presumably the observations are independent
The authors seem to be interpreting confidence intervals as a Bayesian construct, as opposed to the classical frequentist definition. Frequentist confidence intervals can be problematic, especially when considering ratios of poorly-characterised quantities, as here. One can obtain 95% confidence intervals that cover the whole real line, for example.
The algorithm on p7, lines 219 et seq, seems to have little value. It is very difficult to understand and of doubtful value.
tables 1 and 2 it would be nice to see at least one histogram of length of a CI, I would expect the length to be strongly skewed in most cases, does this occur?
Author Response
Response to the Referee
Title: Confidence intervals for the ratio of coefficients of variation of Weibull distributions with applied to wind speed data
Authors: Manussaya La-ongkaew, Sa-Aat Niwitpong and Suparat Niwitpong
Dear referee,
We would like to express our gratitude to referees, who have given valuable suggestions and comments towards the improvement of our manuscript. This manuscript was revised according to the suggestions of referees. Here is one-by-one response to your comments as below. Also, the English of this paper is edited by native speaker.
Reviewer 3:
The English is uniformly poor and must be improved by careful editing by a native speaker. Almost every sentence contains a grammatical error. The meaning is usually (but not always) clear but the poorly structured sentences give a very amateurish feel to the manuscript.
- The authors seem to confuse global warming with the catalytic degradation of the ozone layer: these are very different processes and the authors seem to be quite poorly informed about both of them. The long dense paragraphs on pages 2 and 3 are out of place in an introduction.
Response: Thank you for pointing out our mistake. We already removed it.
- It is not clear to me in what way the CV is superior as an inferential tool to the variance and the authors do not make this clear. I would expect a self-contained discussion of this issue. Why not consider a ratio of variances which would have a notional Fisher distribution]? What exactly goes wrong by this that considering ratios of CVs ameliorates?
Response: To better understand, the coefficient of variation (CV) helps to measure the degree of consistency in distribution of the data sets, unlike variance and standard deviation, it does not consider the measurement unit of the original data. For comparison the dispersion of multiple data in different units or very different means, the CV is useful component or alternative to the variance and standard deviation.
This research is interested in comparing the dispersion of wind speed in two areas. the difference and ratio parameter can be applied. However, if the coefficient of variation of each population is small. The difference will vary directly as two CVs, this means they will also be less. As a result, the conclusions are not as clear as they should be. Therefore, the ratio would be more appropriate than the difference.
- p2line104 presumably the observations are independent
Response: Thank you for your suggestion. We already added it.
- The authors seem to be interpreting confidence intervals as a Bayesian construct, as opposed to the classical frequentist definition. Frequentist confidence intervals can be problematic, especially when considering ratios of poorly-characterised quantities, as here. One can obtain 95% confidence intervals that cover the whole real line, for example.
Response: From our literature reviews, the Bayesian estimation is more efficient than the classic, which is maximum likelihood estimation. For our simulation results, Bayesian based on non-informative performs poorly for small ratios. However, the Bayesian based on gamma prior also performed well in this case, which considered coverage probability and expected length.
- The algorithm on p7, lines 219 et seq, seems to have little value. It is very difficult to understand and of doubtful value.
Response: Based on the research of Khan and Ahmed, this code is used to generated the MCMC sample from posterior density function of Weibull distribution, which the prior distribution of shape and scale parameters are uniform. It must be used together with the JAGs package in R programming. We changed ‘lambda’ to ‘theta’, and reported that theta is a transformation of a scale parameter in another form.
- Tables 1 and 2 it would be nice to see at least one histogram of length of a CI, I would expect the length to be strongly skewed in most cases, does this occur?
Response: Thank you for your suggestion. We provided the histogram of length of a CI in the revised manuscript.

Round 2
Reviewer 1 Report
The authors considered all my comments and suggestions.
The manuscript may be accepted.
Author Response
Dear referee,
We would like to express our gratitude to referee, who have given valuable suggestions and comments towards the improvement of our manuscript. This manuscript was revised according to the suggestions of referees.
Reviewer 1:
The authors considered all my comments and suggestions. The manuscript may be accepted.
Response: Thank you for your suggestions. We would like to inform you that the title was changed from “Confidence intervals for the ratio of coefficients of variation of Weibull distributions with applied to wind speed data” to “Estimation of the confidence interval for the ratio of the coefficients of variation of two Weibull distributions and its application to wind speed data”
Reviewer 2 Report
This version has been significantly improved. The authors have answered most of the comments. I suggest the authors cite the following references, which are highly related to generalized inference and Bayesian approaches for Weibull distribution.
1. Wang B, Yu K, Jones M. Inference under progressively type II right-censored sampling for certain lifetime distributions. Technometrics 2010;52(4):453–60.
https://doi.org/10.1198/TECH.2010.08210
2. Zhang L, Xu A, An L, Li M. Bayesian inference of system reliability for multicomponent stress-strength model under Marshall-Olkin Weibull distribution. Systems. 2022, 10(6):196. https://doi.org/10.3390/systems10060196
3. Zhuang L, Xu A, Wang B, Xue Y, Zhang S. Data analysis of progressive‐stress accelerated life tests with group effects. Quality Technology & Quantitative Management. 2022.
https://doi.org/10.1080/16843703.2022.2147690
Author Response
Dear referee,
We would like to express our gratitude to referees, who have given valuable suggestions and comments towards the improvement of our manuscript. This manuscript was revised according to the suggestions of referees. Here is one-by-one response to your comments as below.
Reviewer 2:
This version has been significantly improved. The authors have answered most of the comments. I suggest the authors cite the following references, which are highly related to generalized inference and Bayesian approaches for Weibull distribution.
Wang B, Yu K, Jones M. Inference under progressively type II right-censored sampling for certain lifetime distributions. Technometrics 2010;52(4):453–60.
https://doi.org/10.1198/TECH.2010.08210
Zhang L, Xu A, An L, Li M. Bayesian inference of system reliability for multicomponent stress-strength model under Marshall-Olkin Weibull distribution. Systems. 2022, 10(6):196. https://doi.org/10.3390/systems10060196
Zhuang L, Xu A, Wang B, Xue Y, Zhang S. Data analysis of progressive‐stress accelerated life tests with group effects. Quality Technology & Quantitative Management. 2022.
https://doi.org/10.1080/16843703.2022.2147690
Response: Thank you for your suggestion. We already added the related works in section “Introduction”. We would like to inform you that the title was changed from “Confidence intervals for the ratio of coefficients of variation of Weibull distributions with applied to wind speed data” to “Estimation of the confidence interval for the ratio of the coefficients of variation of two Weibull distributions and its application to wind speed data”
Reviewer 3 Report
see comment for editor, there is a technical problem with the submission
Author Response
Reviewer 3:
The English is uniformly poor and must be improved by careful editing by a native speaker. Almost every sentence contains a grammatical error. The meaning is usually (but not always) clear but the poorly structured sentences give a very amateurish feel to the manuscript.
Response: The revised manuscript has been successfully improved and edited by native speakers. We would like to inform you that the title was changed from “Confidence intervals for the ratio of coefficients of variation of Weibull distributions with applied to wind speed data” to “Estimation of the confidence interval for the ratio of the coefficients of variation of two Weibull distributions and its application to wind speed data”
- The authors seem to confuse global warming with the catalytic degradation of the ozone layer: these are very different processes and the authors seem to be quite poorly informed about both of them. The long dense paragraphs on pages 2 and 3 are out of place in an introduction.
Response: Thank you for pointing out our mistake. We already removed it.
- It is not clear to me in what way the CV is superior as an inferential tool to the variance and the authors do not make this clear. I would expect a self-contained discussion of this issue. Why not consider a ratio of variances which would have a notional Fisher distribution]? What exactly goes wrong by this that considering ratios of CVs ameliorates?
Response: To better understand, the coefficient of variation (CV) helps to measure the degree of consistency in distribution of the data sets, unlike variance and standard deviation, it does not consider the measurement unit of the original data. For comparison the dispersion of multiple data in different units or very different means, the CV is useful component or alternative to the variance and standard deviation.
This research is interested in comparing the dispersion of wind speed in two areas. the difference and ratio parameter can be applied. However, if the coefficient of variation of each population is small. The difference will vary directly as two CVs, this means they will also be less. As a result, the conclusions are not as clear as they should be. Therefore, the ratio would be more appropriate than the difference.
- p2line104 presumably the observations are independent
Response: Thank you for your suggestion. We already added it.
- The authors seem to be interpreting confidence intervals as a Bayesian construct, as opposed to the classical frequentist definition. Frequentist confidence intervals can be problematic, especially when considering ratios of poorly-characterised quantities, as here. One can obtain 95% confidence intervals that cover the whole real line, for example.
Response: From our literature reviews, the Bayesian estimation is more efficient than the classic, which is maximum likelihood estimation. For our simulation results, Bayesian based on non-informative performs poorly for small ratios. However, the Bayesian based on gamma prior also performed well in this case, which considered coverage probability and expected length.
- The algorithm on p7, lines 219 et seq, seems to have little value. It is very difficult to understand and of doubtful value.
Response: Based on the research of Khan and Ahmed, this code is used to generated the MCMC sample from posterior density function of Weibull distribution, which the prior distribution of shape and scale parameters are uniform. It must be used together with the JAGs package in R programming. We changed ‘lambda’ to ‘theta’, and reported that theta is a transformation of a scale parameter in another form.
- Tables 1 and 2 it would be nice to see at least one histogram of length of a CI, I would expect the length to be strongly skewed in most cases, does this occur?
Response: Thank you for your suggestion. We provided the histogram of length of a CI in the revised manuscript.

Round 3
Reviewer 3 Report
The authors submit a convincing rebuttal and have altered the manuscript appropriately. The work is substantially improved, I detected only minor grammar infelicities. I was fascinated by Figure 2 showing the lengths of the confidence intervals and was surprised to see the large differences between ABCD and EF, it would be interesting to see how the authors interpret this. Also could we reconfigure this to be an ECDF with six lines on it so all could be on the same axes?
Author Response
Dear referee,
We would like to express our gratitude to referee, who have given valuable suggestions and comments towards the improvement of our manuscript. This manuscript was revised according to the suggestions of referees. Here is one-by-one response to your comments as below.
Reviewer 3:
The authors submit a convincing rebuttal and have altered the manuscript appropriately. The work is substantially improved, I detected only minor grammar infelicities. I was fascinated by Figure 2 showing the lengths of the confidence intervals and was surprised to see the large differences between ABCD and EF, it would be interesting to see how the authors interpret this. Also, could we reconfigure this to be an ECDF with six lines on it so all could be on the same axes?
Response: Thank you for pointing out our mistake. Previously, we intended this figure to indicate which confidence interval lengths of the methods is the shortest, that is one criterion for demonstrating that the method is effective. To better understand, we have changed the graph to be in line with the tables shown in the simulation results. They are provided in Figures 1 and 2 for equal and unequal sample sizes, respectively.
